# GeoGS3D: Single-view 3D Reconstruction via Geometric-aware Diffusion Model and Gaussian Splatting

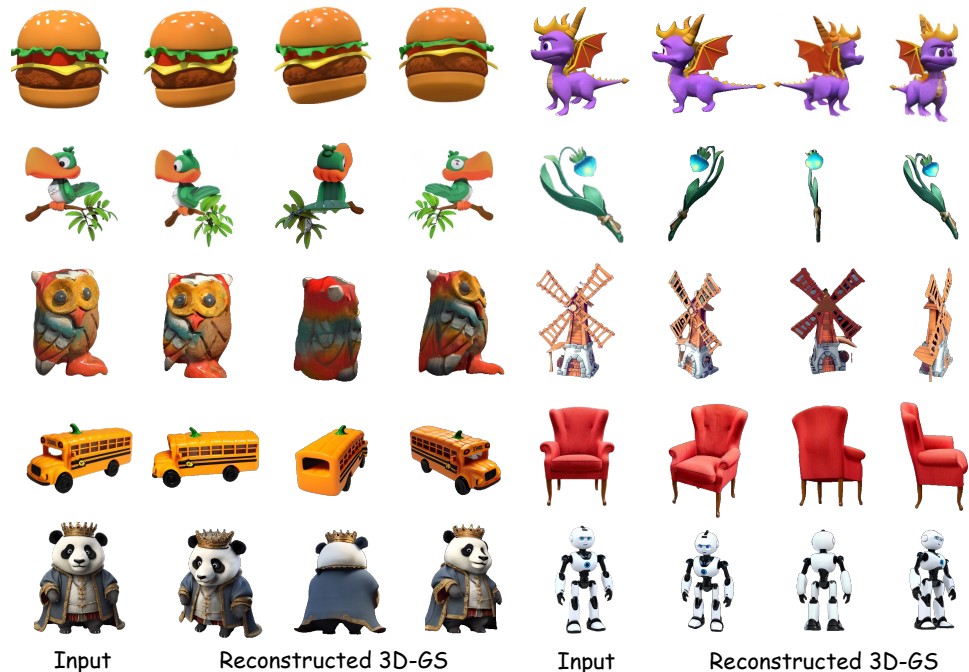

Figure 1: **GeoGS3D for single image to 3D generation:** GeoGS3D can reconstruct 3D content with detailed geometry and accurate appearance from a single image.

## Abstract

We introduce GeoGS3D, a novel two-stage framework for reconstructing detailed 3D objects from single-view images. Inspired by the success of pre-trained 2D diffusion models, our method incorporates an orthogonal plane decomposition mechanism to extract 3D geometric features from the 2D input, facilitating the generation of multi-view consistent images. During the following Gaussian Splatting, these images are fused with epipolar attention, fully utilizing the geometric correlations across views. Moreover, we propose a novel metric, Gaussian Divergence Significance (GDS), to prune unnecessary operations during optimization, significantly accelerating the reconstruction process. Extensive experiments demonstrate that GeoGS3D generates images with high consistency across views and reconstructs high-quality 3D objects, both qualitatively and quantitatively. Further examples can be found at the anonymous website.

## 1 Introduction

Single-view 3D reconstruction aims to recover 3D geometry and appearance of an object from a single RGB image. This task holds immense importance as it allows machines to understand and

interact with the real 3D world, enabling various applications in virtual reality (VR), augmented reality (AR) (Kopf et al., 2020; Li et al., 2023) and robotics (Wang et al., 2022). Despite broad applications of the task, it remains highly challenging due to the inherent complexity of inferring 3D structures from a single 2D image.

Poineering works in multi-view diffusion models have made strides in improving 3D reconstruction by finetuning pre-trained image or video diffusion models on 3D datasets to enable multi-view synthesis (Liu et al., 2023c; Shi et al., 2023; Kwak et al., 2024; Huang et al., 2024; Voleti et al., 2024; Long et al., 2024). However, maintaining multi-view consistency and handling objects with complex geometries remain difficult. Another line of research (Hong et al., 2023; Tang et al., 2024a; Xu et al., 2024c;b) proposes generalizable reconstruction models, generating 3D representation from one or few views in a feed-forward process, but these models usually demand substantial computational resources. For instance, LGM (Tang et al., 2024a) requires training on 32 NVIDIA A100 (80G) for 4 days.

Meanwhile, 3D Gaussian Splatting (Kerbl et al., 2023) has emerged as a promising technique, offering high-quality rendering with competitive training and inference time as it combines the benefits of neural network-based optimization and explicit, structured data storage. Hence, it is desirable to explore Gaussian Splatting for efficient 3D reconstruction. However, current methods (Szymanowicz et al., 2023a; Tang et al., 2023a) using 3D Gaussian Splatting often feed a single image into the model, ignoring the spatial correspondence of multiple views. Additionally, we observe that the original implementation of Gaussian Splatting neglects the distance between 3D Gaussians, causing many unnecessary split and clone operations.

To address these limitations, we propose **GeoGS3D**, a novel two-stage framework for single-image 3D reconstruction. Our approach consists of a geometric-aware multi-view generation stage, followed by an accelerated 3D reconstruction stage. In the generation stage, we aim to synthesize 3D-aware images that maintain multi-view consistency. To achieve the goal, 3D features are extracted as geometric conditions by decoupling the orthogonal planes, while the semantic condition is obtained with the CLIP (Radford et al., 2021) encoder. These conditions, together with the input image are then fed into the diffusion model (Rombach et al., 2022). In the reconstruction stage, we introduce epipolar attention to fuse the generated views, fully leveraging the geometric correlations between them. Furthermore, we accelerate the optimization process by introducing a novel metric, Gaussian Divergent Significance (GDS), to avoid unnecessary operations during 3D Gaussian Splatting.

Extensive experiments and ablations on Objaverse (Deitke et al., 2023) and Google Scanned Object (Downs et al., 2022) datasets demonstrate that our method is able to produce high-quality 3D objects with strong multi-view consistency and detailed geometry. Our key contributions can be summarized as following:

- We incorporate an orthogonal plane decomposition mechanism with a diffusion model to synthesize multi-view consistent and geometric-aware novel view images.
- In order to take full advantage of the consistent multi-view images, we introduce epipolar attention into the optimization process, allowing for efficient and effective communication between images.
- To accelerate the optimization of Gaussian splatting, we derive a novel metric named Gaussian Divergent Significance (GDS) to prune unnecessary split and clone operations during optimization.

## 2 RELATED WORK

### 2.1 MULTI-VIEW DIFFUSION MODELS

Recent 2D diffusion models (Song et al., 2020a; Ho et al., 2020; Song & Ermon, 2019; Song et al., 2020b; Xing et al., 2023c;a;b) make impressive advances in generating images from various conditions. As an intermediate 3D representation, the generation of multi-view images using diffusion models has been explored. The advantage of multi-view images is that they are batched 2D projections and can be directly processed by existing image diffusion models with minor changes. Zero-123 (Liu et al., 2023c) injects camera view as an extra condition to the diffusion model for generating images from different perspectives. Additionally, Shi et al. (2023) proposes replacing self-attention with

multi-view attention in the UNet to produce multi-view consistent images. Other works (Chan et al., 2023; Liu et al., 2023a; Szymanowicz et al., 2023b; Yang et al., 2024; Tseng et al., 2023; Sargent et al., 2023) follow a similar approach to make diffusion model 3D-aware and enhance generation consistency. However, most of these works are not designed for reconstruction tasks and still rely on SDS loss to derive 3D content. Recent efforts such as SyncDreamer (Liu et al., 2023d) and Wonder3D (Long et al., 2024) generate multi-view consistent 2D representations, subsequently applying reconstruction methods to obtain 3D content. Despite these advancements, the resulting 3D outputs often lack comprehensive geometric information. In contrast, our method generates more consistent views with detailed geometric information, facilitating high-quality 3D content reconstruction in terms of texture and geometry.

## 2.2 3D RECONSTRUCTION PIPELINES

Researchers have explored directly training diffusion models on 3D representations (Jun & Nichol, 2023; Müller et al., 2023; Shue et al., 2023). However, they require exhaustive 3D data and computation resources and are also limited to category-level shape generation with simple textures. Other works (Poole et al., 2022; Melas-Kyriazi et al., 2023; Kwak et al., 2024; Sun et al., 2023; Tang et al., 2023a;b; Wang et al., 2023; Zhang et al., 2023) proposed to lift 2D diffusion models for 3D generation via Score Distillation Sampling (SDS), optimizing various 3D representations including NeRF, mesh, SDF and Gaussian Splatting. However, these approaches often face challenges such as prolonged optimization times, the Janus problem, and over-saturated colors. More recent works (Xu et al., 2023; Szymanowicz et al., 2023b; Siddiqui et al., 2024; Charatan et al., 2024; Xu et al., 2024a; Tochilkin et al., 2024; Li et al., 2024) train large reconstruction models to directly map sparse views to 3D representations. Notably, LRM (Hong et al., 2023), Instant3D (Li et al., 2023), and GRM (Xu et al., 2024c) utilize large transformers to predict triplanes from single or sparse views. TriplaneGaussian (Zou et al., 2024) and LGM (Tang et al., 2024a) instead map sparse views into more memory-efficient 3D Gaussian Splatting, allowing for much higher resolution in supervision. Our work aligns with Liu et al. (2023b;a); Tang et al. (2024b); Wen et al. (2024), where we first multi-view diffusion model to generate consistent images, and then utilize reconstruction methods to obtain the corresponding 3D model.

## 3 METHODOLOGY

The overview of GeoGS3D is shown in Figure 2. Starting from an input image, our geometric-aware diffusion model first generates multi-view images sequentially in the generation stage (refer to Section 3.2). Subsequently, in the reconstruction stage, epipolar attention is incorporated into the Gaussian Splatting process to reconstruct the high-quality 3D objects (refer to Section 3.3). Additionally, we introduce a metric to accelerate the adaptive density control during Gaussian Splatting (refer to Section 3.4).

### 3.1 PRELIMINARIES

**Notations** Given an input image $\mathcal{I}_0$, our method aims to generate multiview-consistent images $\{\mathcal{I}_i\}_{i=1}^N$ and then reconstruct high-quality 3D Gaussians $\mathcal{G}$ with them. The multi-view images are obtained through a diffusion model conditioned on the input image $\mathcal{I}_0$ and a set of corresponding relative camera pose change $\{\Delta\pi_i\}_{i=1}^N$ during the generation stage. In the following reconstruction stage, the 3D Gaussians $\mathcal{G}$ are then optimized with these multi-view images through an accelerated Gaussian Splatting.

**3D Gaussian Splatting** (Kerbl et al., 2023) is a learning-based rasterization technique for 3D scene reconstruction and novel view synthesis. Each Gaussian element is defined with a position (mean) $\boldsymbol{\mu}$, a full 3D covariance matrix $\boldsymbol{\Sigma}$, color $c$, and opacity $\sigma$. The Gaussian function $G(x)$ can be formulated as:

$$G(x) = exp(-\frac{1}{2}(\boldsymbol{x} - \boldsymbol{\mu})^T \boldsymbol{\Sigma}^{-1}(\boldsymbol{x} - \boldsymbol{\mu})). \tag{1}$$

To ensure the positive semi-definiteness of $\boldsymbol{\Sigma}$, the covariance matrix $\boldsymbol{\Sigma}$ can be factorized into a scaling matrix $S$ represented by a 3D-vector $s \in \mathbb{R}^3$ and a rotation matrix $R$ expressed as a quaternion $q \in \mathbb{R}^4$ for the differentiable optimization: $\boldsymbol{\Sigma} = RSS^T R^T$.

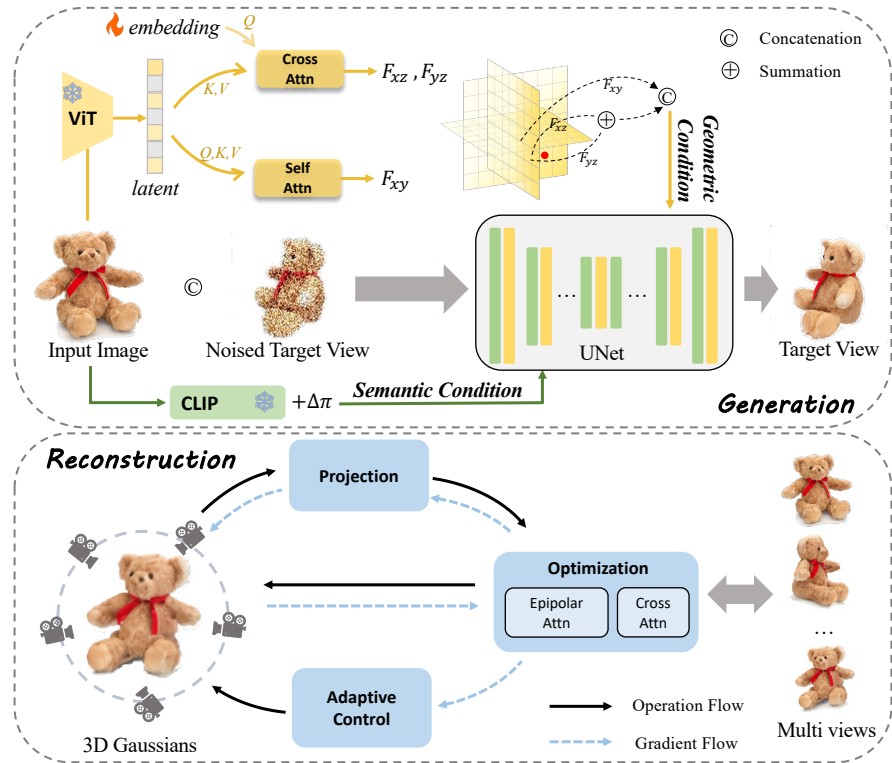

Figure 2: **Overview of our method.** In *generation stage*, we extract 3D features from the single input image by decoupling the orthogonal planes, and feed them into the UNet to generate high-quality multi-view images. In *reconstruction stage*, we leverage the epipolar attention to fuse images with different viewpoints. We further leverage Gaussian Divergent Significance (GDS) to accelerate the adaptive density control during optimization, allowing competitive training and inference time.

The rendering technique of splatting, as initially introduced in Kerbl et al. (2023), is to project the Gaussians onto the camera image planes, which are employed to generate novel view images. Given a viewing transformation $W$, the covariance matrix $\Sigma'$ in camera coordinates is given as: $\Sigma' = JW\Sigma W^T J^T$, where $J$ is the Jacobian matrix of the affine approximation of the projective transformation. After mapping 3D Gaussians to a 2D image space, we count 2D Gaussians that overlap with each pixel and calculate their color $c_i$ and opacity $\sigma_i$ contribution. Specifically, the color of each Gaussian is assigned to every pixel based on the Gaussian representation described in Equation 1. And the opacity controls the influence of each Gaussian. The per-pixel color $\hat{C}$ can be obtained by blending N ordered Gaussians: $\hat{C} = \sum_{i \in N} c_i \sigma_i \prod_{j=1}^{i-1}(1 - \sigma_i)$.

### 3.2 GEOMETRY-AWARE MULTI-VIEW GENERATION

Considering the success of pre-trained diffusion models (Rombach et al., 2022), it is intuitive to finetune them for novel view synthesis under a given camera transformation. However, maintaining multi-view consistency across views remains a substantial challenge. One stream of methods (Woo et al., 2023; Ye et al., 2023) addresses the problem by conditioning on previously generated images, which tends to be susceptible to cumulative errors and reduced processing speeds. Another stream of methods (Liu et al., 2023c;a) solely use the reference image and semantic guidance to generate novel views, but suffer from collapsed geometry and limited fidelity.

We argue that the key lies in fully utilizing the geometric information provided by the input image. However, directly extracting 3D information from a single 2D image is not feasible. Thus, it is imperative to effectively disentangle 3D features from the image plane (*i.e.* $xy$-plane) by decoupling orthogonal planes. We first employ a vision transformer to encode the input image and capture overall correlations in the image, generating high-dimensional latent $\boldsymbol{h}$. Then we leverage two decoders, an

image-plane decoder and an orthogonal-plane decoder, to generate geometric-aware features from the latent. The image-plane decoder reverses the encoding operation, leveraging a self-attention mechanism on the encoder output and converting it into $F_{xy}$. In order to generate orthogonal-plane features while maintaining structural alignment with the image plane, a cross-attention mechanism is employed to decode $yz$ and $xz$ plane features $F_{yz}$ and $F_{xz}$. To facilitate the decoding process across different planes, we introduce a learnable embedding $\boldsymbol{u}$ that supplies additional information for decoupling new planes. The learnable embedding $\boldsymbol{u}$ is first processed through self-attention encoding and then used as a query in a cross-attention mechanism with the encoded image latent $\boldsymbol{h}$. The image features are converted into keys and values for the cross-attention mechanism as following:

$$\texttt{CrossAttn}(\boldsymbol{u},\boldsymbol{h}) = \texttt{SoftMax}\left(\frac{(W^Q\texttt{SelfAttn}(\boldsymbol{u}))(W^K\boldsymbol{h})^T}{\sqrt{d}}\right)(W^V\boldsymbol{h}), \qquad (2)$$

where $W^Q$, $W^K$, and $W^V$ are learnable parameters and $d$ is the scaling coefficient. Finally, the features are combined as geometric conditions:

$$F = F_{xy}\copyright(F_{yz} + F_{xz}), \qquad (3)$$

where $\copyright$ and $+$ are concatenation and summation operations, respectively.

**Training Objective** Similar to previous works (Rombach et al., 2022; Ho et al., 2020), we use a latent diffusion architecture with an encoder $\mathcal{E}$, a denoising UNet $\epsilon_\theta$, and a decoder $\mathcal{D}$. Following Liu et al. (2023c) and Liu et al. (2023d), the input view is channel-concatenated with the noisy target view as the input to UNet. We employ the CLIP image encoder (Radford et al., 2021) for encoding $\mathcal{I}_0$, while the CLIP text encoder (Radford et al., 2021) is utilized for encoding the relative camera pose $\Delta\pi$. The concatenation of their embeddings, denoted as $c(\mathcal{I}_0, \Delta\pi_i)$, forms the semantic condition in the framework. We can learn the network by optimizing the following objective:

$$\mathcal{L}_{mv} = \mathbb{E}_{z\sim\mathcal{E}(\mathcal{I}),t,\epsilon\sim\mathcal{N}(0,1)}\|\epsilon - \epsilon_\theta(z_t, t, c(\mathcal{I}_0, \Delta\pi_i))\|_2^2 \qquad (4)$$

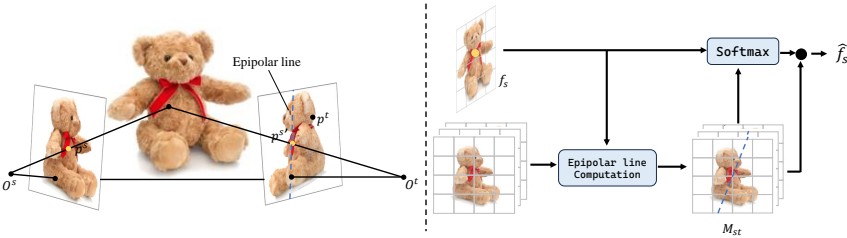

Figure 3: **Illustration of epipolar line and epipolar attention** The epipolar line for a given feature point in one view is the line on which the corresponding feature point in the other view must lie, based on the known geometric transformation.

### 3.3 RECONSTRUCTION WITH EPIPOLAR ATTENTION

During the reconstruction stage, we target to exploit the synthesized consistent multi-view images for restoring high-quality 3D objects. However, relying solely on cross-attention to communicate between images of multiple viewpoints is insufficient. Therefore, we propose epipolar attention to allow association between the features of different views. The epipolar line for a given feature point in one view is the line on which the corresponding feature point in the other view must lie, based on the known geometric relationship between two views. It acts as a constraint to reduce the number of potential pixels in one view that can attend to another view. We present the illustration of epipolar line and epipolar attention in Figure 3.

Consider the intermediate UNet feature $f_s$, we can compute its corresponding epipolar lines $\{l_t\}_{t\neq s}$ on the feature map of all other views $\{f_t\}_{t\neq s}$ (please refer to the Appendix B for the details). Each point $p$ on $f_s$ will only access the features that lie along the camera ray (in other views) as all points in its own views during rendering. We then estimate the weight maps for all positions in $f_s$, stack

these maps, and get the epipolar weight matrix $M_{st}$. Finally, the output of the epipolar attention layer $\hat{f}_s$ can be formulated as:

$$\hat{f}_s = \texttt{SoftMax}\left(\frac{f_s M_{st}^T}{\sqrt{d}}\right) M_{st}. \tag{5}$$

Our proposed epipolar attention mechanism limits the search space for corresponding features across different views, facilitating the efficient and accurate association of features across multiple views. In this way, we effectively reduce the computation cost as well as eliminate potential artifacts.

**Training Objective**    Given the input image $\mathcal{I}_0$ and multi-view images $\{\mathcal{I}_i\}_{i=1}^N$ with relative camera pose change $\{\Delta\pi_i\}_{i=1}^N$, we feed them into the network, and minimize the average reconstruction loss:

$$\mathcal{L}_{rec} = \frac{1}{N}\sum_{s=1}^N \|\mathcal{I}_i - g(f(\mathcal{I}_0), \Delta\pi_i)\|^2, \tag{6}$$

where $g$ is the renderer that maps the set of Gaussians to an image and $f$ is an inverse function that reconstructs the mixture of Gaussians from an image.

The efficiency of our method stems from the idea that it renders the entire image at each training iteration. Therefore, instead of decomposing the results into pixels, we can leverage image-level losses as a whole. In practice, we employ SSIM loss to ensure the structural similarity between ground truth and synthesized images, and LPIPS loss for image quality, *i.e.*

$$\mathcal{L} = \mathcal{L}_{rec} + \lambda_1 \mathcal{L}_{SSIM} + \lambda_2 \mathcal{L}_{LPIPS}, \tag{7}$$

where $\lambda_1$ and $\lambda_2$ are the hyper-parameters of loss weights. Empirically, we set $\lambda_1 = 0.02$ and $\lambda_2 = 0.01$ as default.

## 3.4 Accelerating the Reconstruction

The optimization of Gaussian Splatting is based on successive iterations of rendering and comparing the resulting image to the training views. 3D Gaussians are first initialized from either Structure-from-Motion (SfM) or random sampling. Inevitably, geometry may be incorrectly placed due to the ambiguities of 3D to 2D projection. The optimization process thus needs to be able to adaptively create geometry and also remove geometry (termed as *split* and *clone*) if it is incorrectly positioned.

However, the split and clone operations proposed by the original work (Kerbl et al., 2023) overlook the distance between 3D Gaussians, during the optimization process which significantly slows down the process. We observe that if two Gaussians are close to each other, even if the positional gradients are larger than a threshold, they should not be split or cloned since these Gaussians are updating their positions. Empirically, splitting or cloning these Gaussians has negligible influence on the rendering quality as they are too close to each other. For this reason, we propose Gaussian Divergent Significance (GDS) as a measure of the distance of 3D Gaussians to avoid unnecessary splitting or cloning:

$$\Upsilon_{GDS}(G(\boldsymbol{x}_1), G(\boldsymbol{x}_2)) = \|\boldsymbol{\mu}_1 - \boldsymbol{\mu}_2\|^2 + tr(\boldsymbol{\Sigma}_1 + \boldsymbol{\Sigma}_2 - 2(\boldsymbol{\Sigma}_1^{-1}\boldsymbol{\Sigma}_2\boldsymbol{\Sigma}_1^{-1})^{1/2}), \tag{8}$$

where $\boldsymbol{\mu}_1$, $\boldsymbol{\Sigma}_1$, $\boldsymbol{\mu}_2$, $\boldsymbol{\Sigma}_2$ are the position and covariance matrix of two 3D Gaussians $G(\boldsymbol{x}_1)$ and $G(\boldsymbol{x}_2)$.

To be specific, we only perform the split and clone operations on the 3D Gaussians with large positional gradients and GDS. To avoid the time-consuming process of calculating GDS for every pair of 3D Gaussians, we further propose two strategies. Firstly, for each 3D Gaussian, we locate its closest 3D Gaussian by leveraging the k-nearest neighbor (k-NN) algorithm and calculate their GDS for each pair. As a result, the time complexity is reduced from $O(N^2)$ to $O(N)$. Additionally, as mentioned in Section 3.1, the covariance matrix can be factorized into a scaling matrix $S$ and a rotation matrix $R$: $\boldsymbol{\Sigma} = RSS^T R^T$. We take advantage of the diagonal and orthogonal properties of rotation and scaling matrices to simplify the calculation of Equation 8. Details of GDS will be discussed in the Appendix A.

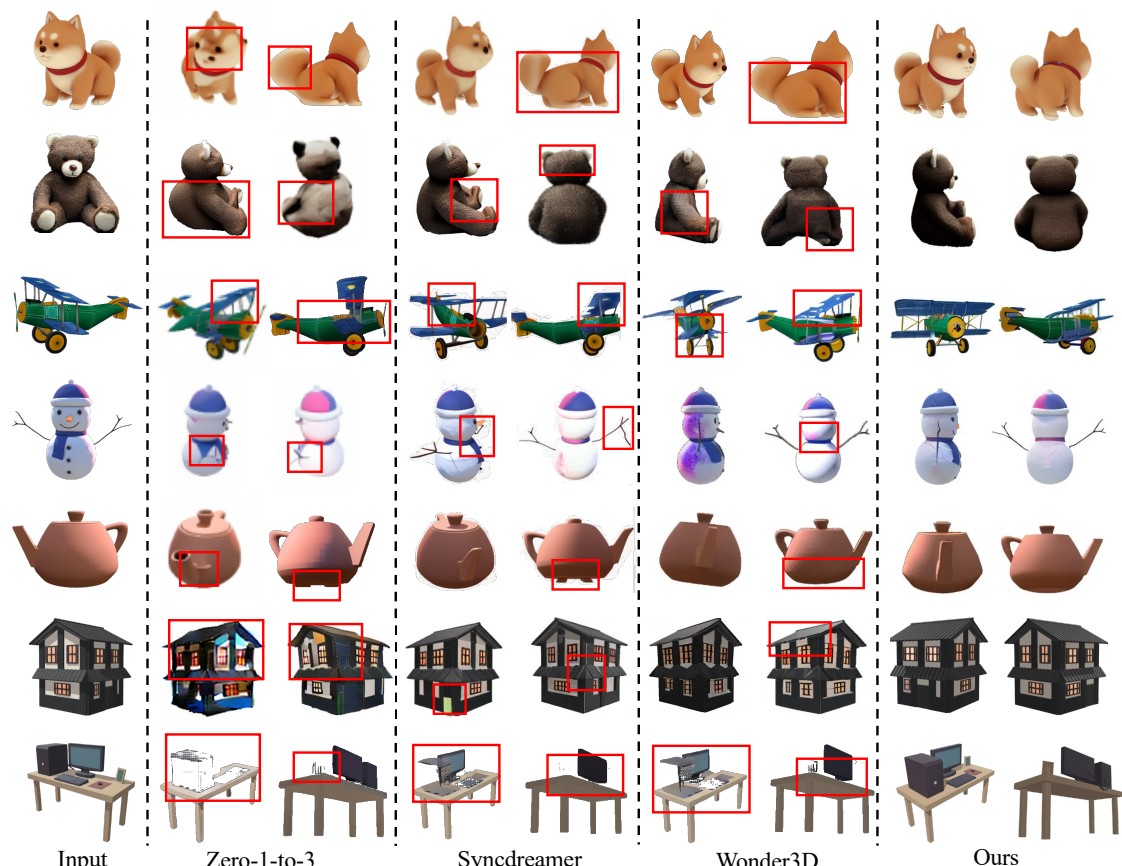

Input Zero-1-to-3 Syncdreamer Wonder3D Ours

Figure 4: **Qualitative comparisons** of generated multi-view images from Objaverse dataset. The artifacts are marked with red boxes. Our method achieves better consistency and visual quality.

## 4 EXPERIMENTS

### 4.1 EXPERIMENTAL PROTOCOLS

**Datasets** We train our multi-view generation model on the recently released Objaverse (Deitke et al., 2023) dataset, which is a large-scale CAD dataset containing 800K high-quality objects. We directly employ the processed rendering data from Zero-1-to-3, which provides 12 random views of each object. During the training, images are resized to $256 \times 256$ resolution.

We evaluate the synthesized multi-view images and reconstructed 3D Gaussian Splatting (3D-GS) on the test split of Objaverse where we randomly select 100 objects beyond the training set. In addition, to test the performance of our model on the out-of-distribution data, we also evaluate the Google Scanned Object (GSO) dataset (Downs et al., 2022), which contains high-quality scanned household items.

**Implementation Details** For the generation stage, we initialize the network with Zero-1-to-3 (Liu et al., 2023c) pre-trained weights for training efficiency. We utilize a Vision Transformer (ViT) model of depth 6 as the reference image encoder and generate an output of size $1024 \times 256$. The decoding process involves two decoders, *i.e.* image plane decoder and orthogonal plane decoder, each with a depth of 3 and outputs a feature map $F \in \mathbb{R}^{128 \times 128 \times 64}$. After the multi-view generation, we directly adopt the implementation of Yang et al. (2022) to select 32 views with the highest perceptual quality score. For the reconstruction stage, the network that maps the input images to the mixtures of Gaussians is architecturally identical to the UNet (Song et al., 2020a). The last layer is replaced with a $1 \times 1$ convolutional layer with 15 output channels. As mentioned in Section 3.3, in order to

allow the network to coordinate and exchange information between views, we add epipolar attention blocks after residual blocks followed by the cross-attention layers. We use the AdamW optimizer with $\beta_1 = 0.9$ and $\beta_2 = 0.999$ with a learning rate of $10^{-4}$.

**Metrics**   We compare generated multi-view images and rendered views from reconstructed 3D-GS with the ground truth frames, in terms of Peak Signal-to-Noise Ratio (PSNR), Structural Similarity Index (SSIM) (Wang et al., 2004), and Learned Perceptual Image Patch Similarity (LPIPS) (Zhang et al., 2018).

**Baselines**   In terms of multi-view generation, we compare our GeoGS3D with Zero-1-to-3 (Liu et al., 2023c), Syncdreamer (Liu et al., 2023d), and Wonder3D (Long et al., 2024). Zero-1-to-3 finetunes Stable Diffusion to achieve novel view synthesis under a given camera pose. Syncdreamer follows a similar intuition but learns the joint distribution of all target views. Wonder3D proposes cross-domain attention to propagate information between normal and image, generating normal maps and color images simultaneously.

For image-to-3D reconstruction, we adopt DreamGaussian (Tang et al., 2023a), LGM (Tang et al., 2024a) and InstantMesh (Xu et al., 2024b). Notably, LGM and InstantMesh also adopt two-stage frameworks to achieve image-to-3D creation, which generates multi-view images and then utilizes large reconstruction models to obtain 3D objects. DreamGaussian initializes geometry and appearance using single-step SDS loss and then extracts a textured mesh with refinement guided by MSE loss.

### 4.2 COMPARISON WITH EXISTING METHODS

**Image to Multi-view**   We first compare the generation stage of our method against recent image-to-multiview generation methods (Liu et al., 2023c;d; Long et al., 2024). The quantitative results are shown in Table 1, and the qualitative results are shown in Figure 4. GeoGS3D surpasses all baseline methods regarding PSNR, LPIPS, and SSIM, indicating it provides sharper and more accurate results. According to the qualitative results, the nearby views synthesized by GeoGS3D are geometrically and semantically similar to the reference view, while the views with large viewpoint change showcase reasonable hallucination. Furthermore, the orthogonal-plane decomposition mechanism enables our model to capture the details of the input image.

Table 1: **Quantitative comparisons** on the quality of generated multi-view images for image to multi-view task.

| Method | Objaverse | | | GSO | | |
|---|---|---|---|---|---|---|
| | PSNR↑ | SSIM↑ | LPIPS↓ | PSNR↑ | SSIM↑ | LPIPS↓ |
| Zero-1-to-3 | 18.68 | 0.883 | 0.189 | 18.37 | 0.877 | 0.212 |
| SyncDreamer | 20.06 | 0.816 | 0.159 | 20.03 | 0.817 | 0.166 |
| Wonder3D | 23.52 | 0.895 | 0.130 | 22.69 | 0.883 | 0.147 |
| GeoGS3D (generation) | **23.97** | **0.921** | **0.113** | **22.98** | **0.899** | **0.146** |

**Image to 3D**   For the single-image 3D reconstruction task, we show the results in Table 2 and Figure 5. Statistically, GeoGS3D outperforms competing approaches by a substantial margin. From the visual results, our method is able to generalize to unseen data. This demonstrates the superiority of GeoGS3D over the current state-of-the-art methods and its capacity to generate high-quality 3D objects even with complex structures.

Table 2: **Quantitative comparisons** on the quality of 3D representation for image-to-3D.

| Method | Objaverse | | | GSO | | |
|---|---|---|---|---|---|---|
| | PSNR↑ | SSIM↑ | LPIPS↓ | PSNR↑ | SSIM↑ | LPIPS↓ |
| DreamGaussian | 19.97 | 0.860 | 0.205 | 20.24 | 0.856 | 0.179 |
| LGM | 20.44 | 0.861 | 0.222 | 18.72 | 0.842 | 0.231 |
| InstantMesh | 21.63 | 0.891 | 0.148 | 20.95 | 0.871 | 0.147 |
| GeoGS3D (reconstruction) | **22.35** | **0.894** | **0.143** | **21.75** | **0.896** | **0.141** |

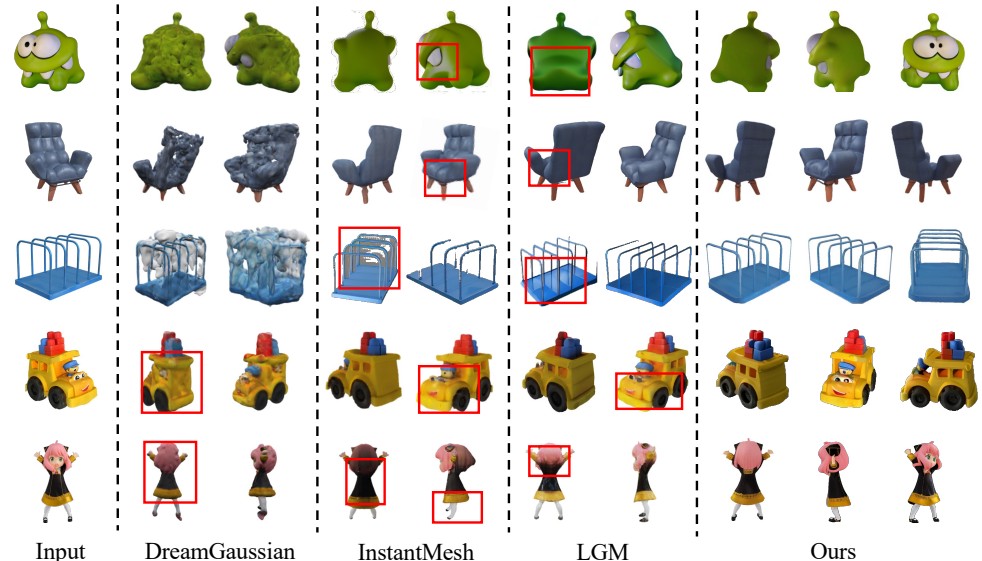

Figure 5: **Qualitative comparisons** for image-to-3D. The first two rows are from the Objaverse dataset, the next two are from the GSO dataset, and the final row is from an in-the-wild image. Our method demonstrates superior performance in terms of both visual fidelity and accuracy compared to existing approaches.

### 4.3  ABLATIONS AND ANALYSES

The ablation studies below are conducted on the GSO dataset (Downs et al., 2022).

**Ablations of multi-view generation**    Our multi-view generation stage mainly consists of geometric and semantic guidance. Removing them respectively or simultaneously gives us four different combinations. As shown in Table 3 and Figure 6, the orthogonal-plane decomposition mechanism contributes the most to the geometric accuracy and consistency, bringing about visual enhancement to a great extent. The semantic guidance further increases the metric score and slightly improves visual consistency.

Table 3: Ablation studies of multi-view generation.

| id | geometric cond. | CLIP embedding | PSNR↑ | SSIM↑ | LPIPS↓ |
|----|----------------|----------------|-------|-------|--------|
| a  | ✓              | ✓              | **22.98** | **0.899** | **0.146** |
| b  | ✓              | ✗              | 20.79 | 0.878 | 0.175 |
| c  | ✗              | ✓              | 18.37 | 0.877 | 0.212 |
| d  | ✗              | ✗              | 17.05 | 0.801 | 0.203 |

**Acceleration of the optimization**    As mentioned in Section 3.4, we propose to use the Gaussian Divergent Significance (GDS) metric to further regularize the split and clone process. Table 4 demonstrates that this strategy has significantly reduced the optimization time while not sacrificing the reconstruction quality. Selecting the threshold carefully, our method leads to at most $15\times$ faster convergence speed when compared with the original split and clone operation proposed in Kerbl et al. (2023).

**Ablations of epipolar attention**    To further validate the effectiveness of the proposed epipolar attention, we conduct ablation studies on it. We still adopt LPIPS to evaluate the quality of reconstruction. As presented in Table 5 and Figure 6, epipolar attention enables accurate association of

---

[1]Here the reconstruction time refers to our second stage, *i.e.* generating 3D Gaussian representations from the multi-view images. Measured on NVIDIA V100 GPU.

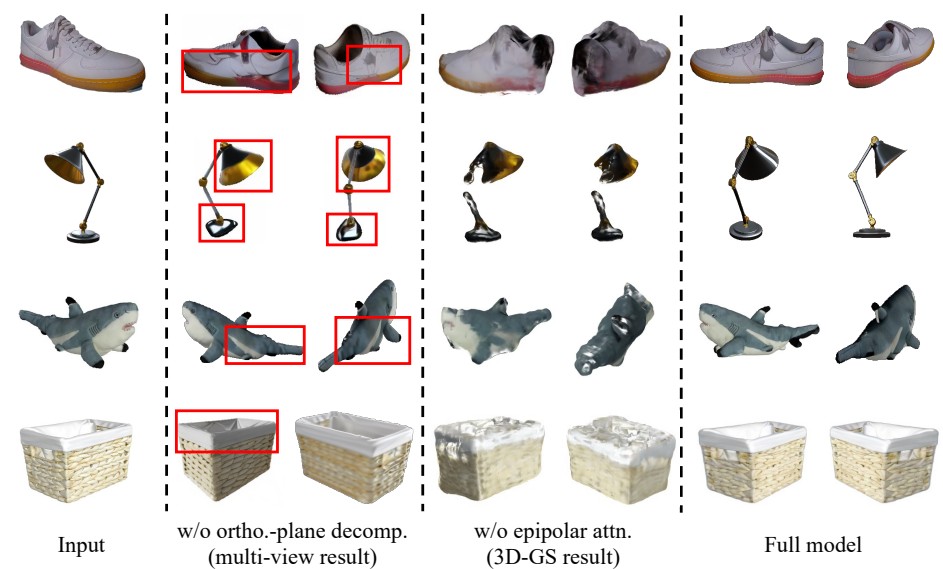

|  Input | w/o ortho.-plane decomp. (multi-view result) | w/o epipolar attn. (3D-GS result) | Full model |

Figure 6: **Qualitative ablations** of our key innovations.

Table 4: **Quantitative results** of ablating GDS metric.

| Threshold | LPIPS↓ | recon. time [1] |
|---|---|---|
| w/o GDS | 0.214 | 15min |
| 0.01 | 0.168 | 93s |
| 0.1 | **0.146** | **55s** |
| 0.5 | 0.147 | 78s |

Table 5: **Ablation studies** of epipolar attention.

| | LPIPS ↓ |
|---|---|
| w/ epipolar attn. | **0.146** |
| w/o epipolar attn. | 0.231 |

features across views, fully utilizing the consistent multi-view images to synthesize high-quality 3D Gaussian Splatting.

### 4.4 LIMITATIONS AND FUTURE WORKS

While GeoGS3D shows promising results in reconstructing 3D objects from single-view images, there are still some limitations that the current framework does not entirely address. First, the number of generated views is fixed in our method. Adaptively generating different numbers of views for objects with different topological symmetries might further reduce the total reconstruction time. Additionally, our current method is restricted to single-object 3D reconstruction. It remains to be extended to complex scenes or multi-object reconstruction in the future.

## 5 CONCLUSIONS

In this work, we propose a two-stage model, GeoGS3D, to reconstruct 3D objects from single-view images. This method first synthesizes consistent and 3D-aware multi-view images via a diffusion model under the guidance of an orthogonal-plane decomposition mechanism. During the following 3D-GS reconstruction, epipolar attention is leveraged to communicate between relating multi-view images. A novel metric, *i.e.* Gaussian Divergent Significance (GDS), is proposed to accelerate optimization. Qualitative and quantitative results show that the proposed method reconstructs 3D Gaussian representations that 1) are consistent in different viewpoints, 2) are high fidelity to the reference image, and 3) display plausible creativity in the unseen areas.

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

## A    DETAILS OF GAUSSIAN DIVERGENT SIGNIFICANCE (GDS)

In this paper, we propose Gaussian Divergent Significance (GDS) as a measure of the distance of 3D Gaussians to avoid unnecessary split or clone operations:

$$\Upsilon_{GDS}(G(\boldsymbol{x}_1), G(\boldsymbol{x}_2)) = \|\boldsymbol{\mu}_1 - \boldsymbol{\mu}_2\|^2 + tr(\boldsymbol{\Sigma}_1 + \boldsymbol{\Sigma}_2 - 2(\boldsymbol{\Sigma}_1^{-1}\boldsymbol{\Sigma}_2\boldsymbol{\Sigma}_1^{-1})^{1/2}). \tag{9}$$

The first term $\|\boldsymbol{\mu}_1 - \boldsymbol{\mu}_2\|^2$ can be expressed as $(\boldsymbol{\mu}_1 - \boldsymbol{\mu}_2)^T(\boldsymbol{\mu}_1 - \boldsymbol{\mu}_2)$. Additionally, the covariance matrix can be factorized into a scaling matrix $S$ and a rotation matrix $R$: $\boldsymbol{\Sigma} = RSS^T R^T$. Since the scaling matrix is a diagonal matrix and the rotation matrix is orthogonal, we can thus simplify the computation of the second term. Given:

$$\begin{aligned} \boldsymbol{\Sigma}_1 &= R_1 S_1 S_1^T R_1^T \\ \boldsymbol{\Sigma}_2 &= R_2 S_2 S_2^T R_2^T, \end{aligned} \tag{10}$$

we have

$$\begin{aligned} &tr(\boldsymbol{\Sigma}_1 + \boldsymbol{\Sigma}_2 - 2(\boldsymbol{\Sigma}_1^{-1}\boldsymbol{\Sigma}_2\boldsymbol{\Sigma}_1^{-1})^{1/2}) \\ &= tr(\boldsymbol{\Sigma}_1) + tr(\boldsymbol{\Sigma}_2) - tr(2(\boldsymbol{\Sigma}_1^{-1}\boldsymbol{\Sigma}_2\boldsymbol{\Sigma}_1^{-1})^{1/2}) \\ &= tr(R_1 S_1 S_1^T R_1^T) + tr(R_2 S_2 S_2^T R_2^T) - tr(2(\boldsymbol{\Sigma}_1^{-1}\boldsymbol{\Sigma}_2\boldsymbol{\Sigma}_1^{-1})^{1/2}) \\ &= tr(S_1 S_1^T) + tr(S_2 S_2^T) - tr(2(\boldsymbol{\Sigma}_1^{-1}\boldsymbol{\Sigma}_2\boldsymbol{\Sigma}_1^{-1})^{1/2}) \end{aligned} \tag{11}$$

In this way, we can save the computation time for Gaussian Divergent Significance.

## B    DETAILS OF EPIPOLAR LINE

Given the point $p^t$ on the target view image $\mathcal{I}_t$ and the relative camera pose change $\Delta\pi$, we show the computation of the epipolar line at the source view image $\mathcal{I}_s$. The relative pose change $\Delta\pi$ can be represented by the rotation and transformation from view $t$ to $s$ as $R^{t \to s}$ and $T^{t \to s}$. We first project the point $p^t$ onto the source view image plane as $p^{t \to s}$, namely

$$p^{t \to s} = \pi(R^{t \to s}(p^t) + T^{t \to s}), \tag{12}$$

where $\pi$ is the projection function. We also project the camera origin $o^t = [0, 0, 0]^T$ at the target view $t$ onto the source view image plane as $o^{t \to s}$:

$$o^{t \to s} = \pi(R^{t \to s}([0, 0, 0]^T) + T^{t \to s}). \tag{13}$$

Then the epipolar line of the point $p$ on the source view image plane can be formulated as

$$p_{epi} = o^{t \to s} + c(p^{t \to s} - o^{t \to s}) \quad c \in \{-\infty, +\infty\} \in \mathbb{R}. \tag{14}$$

Finally, the distance between a point $p^s$ on the source view image plane and the epipolar line can be computed as

$$d(p_{epi}, p^s) = \frac{\|(p^s - o^{t \to s}) \times (p^{t \to s} - o^{t \to s})\|}{\|p^{t \to s} - o^{t \to s}\|}, \tag{15}$$

where $\times$ and $\|\cdot\|$ indicate vector cross-product and vector norm, respectively. According to the epipolar line, we compute the weight map, where higher pixel values indicate a closer distance to the line

$$m_{st}(p^s) = 1 - \texttt{sigmoid}(60(d(p_{epi}, p^s) - 0.06)) \quad \forall p^s \in \mathcal{I}_s. \tag{16}$$

We use the constant 60 to make the sigmoid function steep and use the constant 0.06 to include the pixels that are close to the epipolar line. After estimating the weight maps for all positions in $f_s$, we stack these maps and reshape them to get the epipolar weight matrix $M_{st}$, which is used to compute the epipolar attention described in the paper.

## C    POTENTIAL SOCIAL IMPACT

Our method mainly focuses on object reconstruction and does not involve the use of human data, ensuring that it does not violate human privacy or raise concerns regarding personal data misuse. However, despite this safeguard, it's imperative to acknowledge potential negative social impacts. We are committed to ensuring that our technology is not misused to generate fake data or facilitate deceptive practices, such as counterfeiting or cheating. Ethical considerations and responsible deployment are paramount in our research and development efforts.

