# OpenReview forum: "GeoGS3D: Single-view 3D Reconstruction via Geometric-aware Diffusion Model and Gaussian Splatting"
_ICLR.cc/2025/Conference — ICLR 2025 Conference Withdrawn Submission_

### Official Review · Reviewer_y5KJ · 2024-10-29

**Soundness:** 1
**Presentation:** 1
**Contribution:** 1
**Rating:** 3
**Confidence:** 4

**Summary:**

This paper introduces a novel framework for generating 3D objects from single images. The proposed approach follows a two-stage process: first, generating multi-view images, and then reconstructing 3D Gaussians from these images. In the generation stage, it employs both geometric and semantic conditioning to guide a diffusion model for generating the target novel views. For the reconstruction stage, epipolar attention is utilized to fuse images from different perspectives, and a GDS is introduced to accelerate the reconstruction process.

**Strengths:**

1. The method is thoroughly evaluated on the Objaverse and GSO datasets, with quantitative results surpassing baseline approaches.

2. The mechanism of epipolar attention is clearly explained, and its application in reconstruction tasks is innovative.

3. The proposed GDS metric significantly reduces optimization time.

**Weaknesses:**

1. The design of the geometric condition is questionable. The goal appears to be the decoupling of features into three orthogonal planes. However, there is no mechanism to ensure that the decoupled features actually lie on orthogonal planes. In other words, there is no rationale for why these design choices could extract **3D** information. It seems to process 2D features solely through 2D attention modules.

2. Section 3.2 lacks an explanation of how the generated images maintain multi-view consistency, or if any module is introduced to enforce it. This is confusing, as the training objective in Equation 4 involves a single target view, and Figure 2 also displays only one target view. Does the model output only one target view per diffusion process, or does it generate multiple views, using cross-attention to ensure consistency (similar to Wonder3D)?

3. Although the calculations involved in epipolar attention are clearly explained, it’s not clear how this module is incorporated into the optimization process. Including an intuitive explanation would improve clarity.

4. The paper could include stronger baseline methods for comparison. For example, in multi-view generation, methods such as Era3D [1] and Zero123++ [2] could be added, and for image-to-3D, methods like TripoSR [3] and SF3D [4] could be relevant. The current baselines do not reflect the state-of-the-art in these fields.

5. The ablation on orthogonal-plane decomposition is unclear. The authors remove geometric guidance to evaluate the effect of the orthogonal-plane decomposition mechanism. However, the correct approach would be to remove the decomposition module itself and use the latent vector from a pre-trained ViT as a condition to isolate the impact of orthogonal-plane decomposition. Since previous works all condition the UNet on the input image feature, removing the condition entirely will certainly degrade performance.

6. The writing is unclear and omits essential details (see above weaknesses and questions sections).

Minor:

1. In Eq.4, the geometric condition is missing.

2. In Eq.6, the subscript $s$ doesn’t appear in the summation, which makes this equation confusing.

[1] Li, Peng, et al. "Era3D: High-Resolution Multiview Diffusion using Efficient Row-wise Attention."

[2] Shi, Ruoxi, et al. "Zero123++: a single image to consistent multi-view diffusion base model."

[3] Tochilkin, Dmitry, et al. "Triposr: Fast 3d object reconstruction from a single image."

[4] Boss, Mark, et al. "Sf3d: Stable fast 3d mesh reconstruction with uv-unwrapping and illumination disentanglement."

**Questions:**

1. Are $F_{yz}$ and $F_{xz}$ generated by the same $u$? Are the image plane decoder and orthogonal plane decoder jointly trained with the multi-view generative model?

2. How many views does the multi-view diffusion generate? How many are generated in total before selecting the 32 views with the highest quality scores?

3. The paper mentions a fixed number of generated views. Are the relative camera poses also fixed?

4. How does the elevation of the input image affect the quality of generated views? For example, would the model fail if the input image is captured from a high elevation angle?

5. How long does the generation stage take, including the selection of 32 views?

**Details Of Ethics Concerns:**

No ethical concerns have been identified.

---

### Official Review · Reviewer_XkGL · 2024-10-31

**Soundness:** 3
**Presentation:** 2
**Contribution:** 1
**Rating:** 3
**Confidence:** 4

**Summary:**

The paper proposes a single view to consistent 3D reconstruction method using geometrically consistent diffusion to produce multiple views for optimizing an explicit 3D reconstruction algorithm (3DGS). The paper proposed epipolar attention in the generation pipeline to achieve better multi-view consistency (over zero123's pretrained weights) and accelerate 3DGS using a so called novel metric Gaussian Divergence Significance.

**Strengths:**

1. Good demonstration of the combination of papers: Zero123, Epipolar attention (EpiDiff) and 3DGS to reconstruct 3D objects from single views. Uses them as components well.
2. Qualitative evaluations look promising
3. Sufficient ablation studies are performed to support the claimed motivation behind components.

**Weaknesses:**

1. Lacks novelty. The epipolar attention mechanism is too similar to the recent paper: EpiDiff (CVPR 2024; 10 citations; [Link to paper](https://openaccess.thecvf.com/content/CVPR2024/papers/Huang_EpiDiff_Enhancing_Multi-View_Synthesis_via_Localized_Epipolar-Constrained_Diffusion_CVPR_2024_paper.pdf))

2. Missing direct reference & baseline: EpiDiff (from the point above) is not cited or compared against as a methodology. In my opinion, this is a directly related work and should ideally have been a baseline at the very least.

3. The GDS metric claim can not be called novel by any means. It is just the Wasserstein distance between two Guassians! Giving an existing distance measure a new name doesn't count as novelty. Maybe the authors should re-consider wording their strategy (as novel) for not pruning/splitting Gaussians based on a Wasserstein-distance threshold.

4. Minor detail: The authors claim that GDS reduces the time complexity from O(N^2) to O(N) by not finding distances for all pairs using kNN. This is not a valid claim. Imagine a super noisy & compact scene. Here every Gaussian is close to each other due to compactness and the noise leads to large gradients - This will include all Gaussians again into consideration. Note that big-O notation is about worst case complexity and not average run-time complexity.

5. Table captions are not the most informative. For ex - the authors could improve table 3 by specifying what variants a, b, c and d are. It's hard to jump around the paper back and forth and retain the numbers at the same time.

6. Why is GDS' formulation (a major claim) in the appendix? Similarly for epipolar attention (appendix B)? These should be a part of the main text since they are the main claims.

7. No in-the-wild object reconstructions are shown. Prior literature and direct usage component: Zero123 did that.

**Questions:**

1. My main concern is the missing reference from Weaknesses (pt 1 and 2) and how is it different from the current method's epipolar attention mechanism as it is claimed to be new.
2. I would suggest the authors to reduce their metric novelty claim (Weaknesses pt 3).
3. Points 4 to 6 in weaknesses (technical details + writing) should be addressed.

---

### Official Review · Reviewer_1CRE · 2024-11-02

**Soundness:** 3
**Presentation:** 2
**Contribution:** 3
**Rating:** 3
**Confidence:** 4

**Summary:**

GeoGS3D introduces a two-stage framework for reconstructing detailed 3D objects from single-view image. In generation stage, the framework innovatively replaces traditional single-view feature with 3D geometric feature, facilitating the generation of multi-view consistent images. In reconstruction stage,  the authors implement Gaussian Divergent Significance (GDS), which accelerates the optimization process by eliminating unnecessary operations during 3D Gaussian Splatting. Experiments demonstrate that GeoGS3D generates images with high consistency across views and reconstructs high-quality 3D objects.

**Strengths:**

- The two-stage method presented in this paper performs remarkably well both in quantitative metrics and visual quality.
- The proposed GSD metric significantly improves the reconstruction speed of GS while enhancing the quality of results.

**Weaknesses:**

- In Line 320, the complexity of KNN is not $O(N)$. For each point, we need to query its nearest neighbors using KD-tree: $O(log N)$. Therefore, the total time complexity is: $N * O(log N) = O(N log N)$
- In Line 221, the authors mention a learnable embedding $u$, however, this element is absent from Figure 2, creating confusion in the presentation. The authors should provide a more explicit explanation of how the output from $CrossAttn(u, h)$ is transformed into the triplane representations $(F_{xy}, F_{yz}, F_{xz})$.
- The paper lacks proper citations for epipolar attention, despite numerous prior works implementing this technique.
  - Generalizable Patch-Based Neural Rendering
  - pixelSplat: 3D Gaussian Splats from Image Pairs for Scalable Generalizable 3D Reconstruction
  - EpiDiff: Enhancing Multi-View Synthesis via Localized Epipolar-Constrained Diffusion

**Questions:**

- In Figure 2, the authors utilize the standard 3D Gaussian Splatting, optimizing Gaussian points attributes through cloning and splitting for individual scenes. However, Line 377 describes using Unet and multiview images to output a fixed number of Gaussian points, which doesn't include point quantity control (similar to LGM). These two processes cannot exist in the same stage. So how is the point splitting and cloning work in this process, and how is the GDS works to  avoid unnecessary operation in splitting and cloning?

- Is the GDS metric a universal acceleration solution, or is it more suitable for object-level reconstruction?

---

### Official Review · Reviewer_RMJz · 2024-11-03

**Soundness:** 3
**Presentation:** 1
**Contribution:** 2
**Rating:** 5
**Confidence:** 4

**Summary:**

The paper proposes a two-stage approach for single-view 3D object reconstruction via multi-view generation with a diffusion model first, followed by fitting of 3D Gaussians to the generated views to obtain the final 3D reconstruction. Compared to previous works, the authors propose a specific network architecture based on orthogonal plane decomposition to encode the image into a conditioning for the multi-view diffusion model. Furthermore, for the second 3D reconstruction stage, the authors claim to introduce epipolar attention into the optimization process and define an additional heuristic for the adaptive density control of 3DGS for faster fitting. An evaluation on Objaverse and Google Scanned Objects shows empirical advantages over baselines, both quantitatively and qualitatively. Ablation studies show the effectiveness of the proposed components.

**Strengths:**

- The paper tackles an interesting and very challenging task of single-view 3D object reconstruction.
- The experimental evaluation shows strong performance with advantages over baselines in terms of
  - multi-view generation with a diffusion model (novel views quantitatively closer to ground truth (table 1) and qualitatively more reasonable (figure 4) and
  - 3D reconstruction (quantitatively better metric scores (table 2) and more detailed qualitative results (figure 5).
- The evaluation comprises both synthetic (Objaverse) as well as real (Google Scanned Objects) datasets with the later being out of training distribution.
- Ablation studies w.r.t. the proposed components (geometric conditioning, epipolar attention, and Gaussian Divergent Significance) validate the effectiveness of the contributions.

**Weaknesses:**

- Lack of clarity: I do not understand major parts of the method.
  - In the reconstruction stage, it is not clear to me whether the method uses per object fitting of 3D Gaussians or a network that predicts Gaussians in a feed-forward manner or a combination of both and if so how, as I do not see how the proposed components are compatible.
    - In lines 70ff., authors name Splatter-Image [1] and DreamGaussians [2] as two current 3DGS-based reconstruction methods, i.e., one feed-forward approach for generalizable reconstruction [1] and one per object fitting with score distillation sampling [2].
    - In lines 133ff., the authors say that their work in terms of 3D reconstruction aligns with One-2-3-45 [3] and MVDiffusion++ [4], again one feed-forward approach [3] and one per object fitting [4]. However, both do not use 3D Gaussian Splatting.
    - Figure 2 shows some combination of the usual optimization loop of 3DGS with epipolar and cross attention, which does not provide a detailed explanation of how these are combined (also not in text).
    - Clues suggesting per object fitting:
      - In lines 150ff., the atuhors write " In the following reconstruction stage, the 3D Gaussians G are then optimized with these multi-view images through an accelerated Gaussian Splatting."
      - The use of adaptive density control with the proposed Gaussian Divergent Significance (GDS) as additional heuristic for when to apply split and clone operations. If Gaussians would be predicted in a feed-forward manner, I do not see how these non-differentiable operations could still be used.
      - The reconstruction time comparison in table 4: 15 minutes reconstruction time (without the GDS metric) should be too much for feed-forward prediction.
    - Clues suggesting feed-forward prediction:
      - The use of epipolar and cross attention. In lines 267ff., the authors say that these modules are applied to "the intermediate UNet feature[s]".
      - The training objective in equation 6 includes f as "an inverse function that reconstructs a mixture of Gaussians from an image" (285f.).
      - Lines 375ff.: "For the reconstruction stage, the network that maps the input images to the mixtures of Gaussians is architecturally identical to the UNet [...], we add epipolar attention blocks after residual blocks followed by the cross-attention layers."
  - The geometric conditioning in section 3.2 is missing an explanation of how exactly the features for the three different planes are used by the diffusion model.
    - The subsection about the training objective (lines 234ff.) does not contain the geometric conditioning at all.
  - If the plane features are aggregated via concatenation and summation (equation 3) and then presumably used as conditioning of the diffusion model via cross attention, then It is unclear why this design choice should be effective:
    - There is no enforcement that the "orthogonal plane features" have anything to do with the three orthogonal planes in 3D.
    - If that is the case, F_yz and F_xz should be even equivalent and therefore redundant.
    - Lines 221f.: "The learnable embedding u is first processed through self-attention ...". Self-attention on learnable embeddings is unnecessary.
  - It is unclear how the relative camera poses are encoded by the CLIP text encoder (cf. 238f.).
  - The epipolar attention is claimed to "reduce the computation cost" (line 277), while it is not clear why this weighted attention should be more efficient.
  - In lines 288ff., the authors say that "the efficiency of our method stems from the idea that it renders the entire image at each training iteration...". All 3DGS approaches usually render entire images at once.

- Lack of related work and clarity regarding novelty of reconstruction with epipolar attention:
  - If epipolar attention is used in an architecture (UNet) that predicts Gaussians in a feed-forward manner, then this has been done before, e.g., by pixelSplat [5] and latentSplat [6].
  - If epipolar attention is used by the multi-view diffusion model itself, then this also has been done before, e.g., by Epidiff [7] and MVDiff [8].

- It is not clear whether the evaluation of the 3D reconstruction in table 2 is disentangled from the multi-view generation, i.e., whether they use the same generated images as input or not.
  - If not, that would be an interesting setup in order to better compare the different reconstruction techniques.
- The qualitative comparisons in figures 4 and 6 miss ground truth for reference.

Minor comments:
- The introduction is a bit repetitive w.r.t. the contributions (first in text form (lines 74ff.), then same again as a list (88ff.).
- As 3D Gaussian Splatting is mainly used as it is (except for the additional heuristic in the density control), I do not think that the extensive recap of it in preliminaries is really needed.

References:
- [1] Splatter Image: Ultra-Fast Single-View 3D Reconstruction. CVPR 2024
- [2] DreamGaussian: Generative Gaussian Splatting for Efficient 3D Content Creation. ICLR 2024
- [3] One-2-3-45: Any Single Image to 3D Mesh in 45 Seconds without Per-Shape Optimization. NeurIPS 2023
- [4] MVDiffusion++: A Dense High-resolution Multi-view Diffusion Model for Single or Sparse-view 3D Object Reconstruction. ECCV 2024
- [5] pixelSplat: 3D Gaussian Splats from Image Pairs for Scalable Generalizable 3D Reconstruction. CVPR 2024
- [6] latentSplat: Autoencoding Variational Gaussians for Fast Generalizable 3D Reconstruction. ECCV 2024
- [7] EpiDiff: Enhancing Multi-View Synthesis via Localized Epipolar-Constrained Diffusion. CVPR 2024
- [8] MVDiff: Scalable and Flexible Multi-view Diffusion for 3D Object Reconstruction from Single-View. CVPR 2024

**Questions:**

Please clarify all points w.r.t. lack of clarity in the weaknesses section. To be more precise the questions are:
1. Are you using per-object optimization, feed-forward prediction, or a combination?
2. If it is a combination, how exactly do optimization and feed-forward components interact?
3. How are the epipolar attention and cross attention integrated into the used reconstruction process?
4. How is the geometric conditioning incorporated into the diffusion model's architecture?
5. How do you make use of the fact that the features should correspond to different planes in 3D?
6. What is the exact format or representation of the camera poses as input to the CLIP text encoder?
7. Why is the proposed epipolar attention supposed to be more efficient, if it is a weighted dense attention?
8. Do all reconstruction methods in table 2 use the same set of generated multi-view images as input or not?

---

### Official Review · Reviewer_HXfY · 2024-11-03

**Soundness:** 1
**Presentation:** 2
**Contribution:** 2
**Rating:** 3
**Confidence:** 2

**Summary:**

The manuscript introduces a framework named GeoGS3D, which can reconstruct 3D from a single image. It proposes to perform a decomposed attention mechanism for multi-view image generation. It also uses a reconstruction scheme to lift the multivew to 3D. The result of GeoGS3D was compared with prior works on GSO.

**Strengths:**

-	The problem is of interest to the research community.
-	The paper is easy to follow.

**Weaknesses:**

I am particularly concerned about the scientific contribution and clarity of the paper.

For contribution, my main concern is the claimed “geometry aware multi-view generation”. The paper claims to be able to decompose the multi-view generation into three planes, XY is the image plane, YZ and XZ are 3D planes to enhance the result. However, when looking at the formulation, this is only a variant of multi-head attention, where the QKV are split into three heads. The only difference from multi-head attention is the operation of summation in Eqn 3. The manuscript did not present any evidence of the ability to actually decompose the scene orthogonally.

For the clarity issue, the reconstruction stage is very confusing. Is the reconstruction a feedforward model? If so, what’s the input and output? Why is 3.4 iterative optimization? Is this done after the feedforward model described in 3.3?

Epipolar attention has been proposed in the research community for a long time, especially in the multi-view stereo literature. The manuscript should not claim this as novelty and should provide citation to prior work, such as [a][b][c][d]. Currently, the work sounds like a simple extension of Zero123 and LGM, with the exception of epipolar attention and potentially a final refinement stage. I am not sure if this is significant enough as scientific discovery.

[a] Chen, A., Xu, Z., Zhao, F., Zhang, X., Xiang, F., Yu, J., & Su, H. (2021). Mvsnerf: Fast generalizable radiance field reconstruction from multi-view stereo. In Proceedings of the IEEE/CVF international conference on computer vision (pp. 14124-14133).
[b] Long, X., Lin, C., Wang, P., Komura, T., & Wang, W. (2022, October). Sparseneus: Fast generalizable neural surface reconstruction from sparse views. In European Conference on Computer Vision (pp. 210-227). Cham: Springer Nature Switzerland.
[c] Li, P., Liu, Y., Long, X., Zhang, F., Lin, C., Li, M., ... & Guo, Y. (2024). Era3D: High-Resolution Multiview Diffusion using Efficient Row-wise Attention. arXiv preprint arXiv:2405.11616.
[d] Li, Y., He, X., Jiang, Y., Liu, H., Tao, Y., & Hai, L. (2022, October). MeshFormer: High‐resolution Mesh Segmentation with Graph Transformer. In Computer Graphics Forum (Vol. 41, No. 7, pp. 37-49).

**Questions:**

-	Geometry-aware multi-view

 - How beneficial is the summation operation compared to multi-head attention? Could authors provide an ablation between multi-head attention and the proposed "cat-sum" operation in Eqn3?
 - Can authors provide more evidence to back up the claim that the model learns to decompose 3D? For examples, visualization or ablation to strengthen the claim of the geometric decomposition.

-	Clarification on reconstruction model
 - Could authors provide a clear, step-by-step description of the entire reconstruction process, including how sections 3.3 and 3.4 relate to each other temporally and functionally?

- Novelty about epipolar attention
 - Could authors clarify what are the differences between the proposed epipolar attention and prior work?
 - Could authors provide citation to these approaches?

---

### Note · Authors · 2024-11-15

I have read and agree with the venue's withdrawal policy on behalf of myself and my co-authors.